# Understanding the Role of Input Token Characters in Language Models: How Does Information Loss Affect Performance?

**Ahmed Alajrami**    **Katerina Margatina**    **Nikolaos Aletras**
Department of Computer Science
University of Sheffield, UK
`{ajsalajrami1, k.margatina, n.aletras}@sheffield.ac.uk`

## Abstract

Understanding how and what pre-trained language models (PLMs) learn about language is an open challenge in natural language processing. Previous work has focused on identifying whether they capture semantic and syntactic information, and how the data or the pre-training objective affects their performance. However, to the best of our knowledge, no previous work has specifically examined how information loss in input token characters affects the performance of PLMs. In this study, we address this gap by pre-training language models using small subsets of characters from individual tokens. Surprisingly, we find that pre-training even under extreme settings, i.e. using only one character of each token, the performance retention in standard NLU benchmarks and probing tasks compared to full-token models is high. For instance, a model pre-trained only on single first characters from tokens achieves performance retention of approximately 90% and 77% of the full-token model in SuperGLUE and GLUE tasks, respectively.[1]

## 1 Introduction

The use of pre-trained language models (PLMs), such as BERT (Devlin et al., 2019), T5 (Raffel et al., 2020), GPT-3 (Brown et al., 2020) and many more, has led to significant advances in natural language processing (NLP) tasks, with many achieving state-of-the-art results. However, the exact mechanisms by which these models learn[2] and represent language are still not fully understood.

The pre-training data evidently plays an important role in the downstream performance of

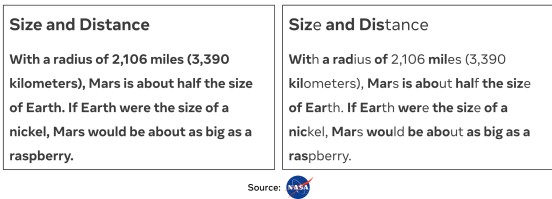

Figure 1: Humans are able to understand natural language text consisting of words with altered letters (e.g., **par**t **b**old, or entiely msig letters). Can language models do the same?

PLMs (Kaplan et al., 2020). Previous studies have mainly focused on exploring how its size affects the performance (Liu et al., 2019b; Yang et al., 2019; Baevski et al., 2019), demonstrating the benefits of using more data together with a larger number of parameters. More recent work has focused on studying the relationship between PLMs performance and the quality and relevance of the pre-training data (Zhang et al., 2020; Raffel et al., 2020; Lee et al., 2022; Micallef et al., 2022; Krishna et al., 2022; Madasu and Srivastava, 2022), showing that high-quality data leads to improved performance. Interestingly, contrary to these findings, other studies showed that language models can still achieve comparable performance even when they are pre-trained using randomly selected character n-grams (Krishna et al., 2021) or pre-trained using synthetic tasks (Wu et al., 2022).

However, no previous studies have so far explored the amount of information that PLMs actually need from individual tokens in the pre-training data. Motivated by this, we pre-train language models by introducing partial information loss in the token sequence. This is achieved by using subsets of characters to represent each token in the pre-training data, i.e. keeping only one, two or three characters at a time (Figure 1). To the best of our knowledge, this study is the first to explore how informative *subsets* of tokens are compared to using

---

[1]Code and models are available here: `https://github.com/aajrami/emnlp2023-token-characters-role`

[2]We may interchangeably use terms such as 'learn', 'comprehend' or 'understand' to refer to the ability of PLMs to understand language. These terms are used in the context of 'natural language understanding' and 'machine reading comprehension' as downstream tasks, highlighting the capabilities of such models to perform well.

*full* tokens in language models pre-training.

Specifically, we draw inspiration from studies in cognitive science and psycholinguistics that have shown that humans possess remarkable flexibility in processing partial word information, such as comprehending text by only reading the first or last characters of each word (McCusker et al., 1981; Rayner et al., 2006; Grainger et al., 2004; Johnson and Eisler, 2012). This intriguing ability raises the question of whether computational models can exhibit similar behavior, bringing forth novel avenues for NLP research. In this paper, we explore the parallels between human reading strategies and language models when presented with partial word information, aiming to uncover common principles and differences underlying both systems.

Our main contributions are as follows:

1. We empirically demonstrate that even when pre-training using one character from the input tokens, PLMs can still effectively learn;

2. We find that the first characters of tokens and consonants are more informative than the last characters and vowels, respectively;

3. We show through probing that specific character positions of tokens assist PLMs in encoding better linguistic information.

## 2   Related Work

Recent efforts in NLP focus on understanding how and what PLMs learn about language including: what information they encode; and how the pre-training objective and data affect what PLMs learn.

**Probing for linguistic information**   A popular area of research is on detecting types of linguistic properties that PLMs encode and to what extent via probing. Typically, a model is trained using the representations of a language model to predict a particular linguistic feature. High accuracy in such a task indicates that the LM effectively encodes that linguistic property (Adi et al., 2016; Hupkes et al., 2018; Conneau et al., 2018).

Probing has shown that PLMs encode linguistic information, such as syntactic and semantic, in their representations (Liu et al., 2019a; Tenney et al., 2019; Lin et al., 2019; Rosa and Mareček, 2019; Hewitt and Manning, 2019; Jawahar et al., 2019; Kim et al., 2019; Vilares et al., 2020; Limisiewicz and Mareček, 2021; Müller-Eberstein et al., 2022; Lasri et al., 2022; Davis et al., 2022).

**Pre-training Objectives**   Another line of work has focused on exploring how pre-training objectives affect the performance of PLMs. Saunshi et al. (2020) demonstrated that PLMs pre-trained on masked language modeling (MLM) perform well in downstream tasks due to their ability to learn semantic dependencies between tokens. However, many different pre-training objectives are effective in downstream tasks (Yang et al., 2019; Clark et al., 2020; Lan et al., 2020; Di Liello et al., 2022). In addition, studies examined how different types of pre-training tasks help PLMs learn, showing that even objectives that are not intuitive for humans, such as tokens grouped in arbitrary random classes, can still help PLMs achieve good performance on downstream tasks and acquire linguistic information (Yamaguchi et al., 2021; Alajrami and Aletras, 2022). Furthermore, a recent study by Yamaguchi et al. (2023) has shown that the performance of PLMs on downstream tasks is significantly influenced by the pre-training objective complexity.

**Pre-training Data**   Closer to our work are studies on investigating the learning process of PLMs by pre-training them on synthetic data. Sinha et al. (2021) found that PLMs pre-trained on sentences with shuffled word order still perform well on downstream tasks, likely due to their ability to capture higher-order co-occurrence statistics. Krishna et al. (2021) used a corpus of randomly selected character n-grams to pre-train models on synthetic summarization tasks. They found that the performance of these models was nearly equivalent to models pre-trained on real data. Ri and Tsuruoka (2022) demonstrated that pre-training on artificial languages that mimic the structural properties of natural language can still result in knowledge transferable to natural language. Even when pre-trained on non-linguistic data such as music or code, language models can still perform well on natural language downstream tasks (Papadimitriou and Jurafsky, 2020; Chiang and Lee, 2020). Similarly, hashing semantically unrelated tokens together in the same embedding results in negligible performance loss (Xue and Aletras, 2022).

Unlike previous work, we explore how much information loss affects the performance of PLMs inspired by human reading comprehension (Fry, 1968; McCusker et al., 1981; Rayner et al., 2006; Grainger et al., 2004; Johnson and Eisler, 2012).

## 3 Models

Various studies in cognitive science have attempted to measure the ability of humans to process words and understand sentences when retaining only some of the characters and their relative position in each word while others are masked, i.e. orthographic priming (Fry, 1968; McCusker et al., 1981; Grainger et al., 2004; Johnson and Eisler, 2012).

Inspired by this, we aim to explore how much information PLMs need from individual tokens by pre-training using only a small subset of characters in each token. We experiment using three different input token subsets (i.e. one, two and three characters) and three relative positions of characters in the token (i.e. first, middle and last). We also experiment with retaining only consonant or vowel characters from each input token. Table 1 shows an overview of the inputs to different models.

### 3.1 Input

**Single Character** First, we experiment with pre-training under extreme settings where it would be nearly impossible for humans to guess the constituent tokens of a sentence or make associations between the input and labels from a downstream task. For this purpose, we only retain a single character from each token:

- **First Character (F):** we use only the first character of each token in the pre-training data as input. For example, 'cat' and 'computer' are both represented by the same token 'c'.

- **Middle Character (M):** we only keep the middle character of each token: $mid = token\left[\left\lfloor\frac{len(token)}{2}\right\rfloor\right]$.

- **Last Character (L):** the last character of each token is retained.

**Two Characters** Further, we pre-train models by increasing the available information using only a maximum of two characters in each token: (1) the first and last (**FL**); (2) the first two (**FF**); and (3) the last two characters (**LL**). These models also consider all tokens consisting of a single character.

**Three Characters** In addition, we use a maximum of three characters from each token as input: (1) the first, middle and last (**FML**); (2) the first three (**FFF**); and (3) the last three (**LLL**). These models also use all tokens consisting of one or two characters.

**Vowels (V)** We further expand our investigation by pre-training models where we exclusively retain the vowel characters (if they exist) from each input token. Through this approach, we explore the capacity of PLMs to acquire and preserve meaningful information solely based on vowel characters.

**Consonants (C)** Finally, to comprehensively explore the impact of different character subsets on pre-training, we also delve into the pre-training of models by exclusively utilizing the consonant characters from each input token.

### 3.2 Architecture and Pre-training

Our main goal is to measure the performance retention of different inputs compared to using a full token. For each different input type, we pre-train a BERT-BASE (Devlin et al., 2019) model by modifying the embedding matrix according to the various vocabularies.[3] We experiment with two pre-training MLM tasks:

**Predicting masked character subsets:** First, we only predict the character subsets of each masked token. For example, given the masked tokens 'phone' and 'photo', the model FF should predict 'ph' for both tokens while the model FL should predict 'pe' and 'po' respectively.

**Predicting the original full token:** We also experiment with a more informative pre-training task by predicting the original full masked token. Note that the input to these models is the character subsets, e.g. F, LL, FML. The results of fine-tuning the models pre-trained using this task on GLUE and SuperGLUE benchmarks are similar to predicting masked characters and can be found in Appendix D and Appendix E respectively.

## 4 Experimental Setup

### 4.1 FULL TOKEN Model

We pre-train a full token model (**Token**) using MLM. This model is used as a reference to compare the performance of our models that use character subsets as input.

### 4.2 Implementation Details

We implement all of our models using PyTorch (Paszke et al., 2019) and the Transformers library (Wolf et al., 2020). We pre-train each model for

---

[3]Exhaustively testing PLMs of different sizes and types is out of the scope of this paper as well as computationally prohibitive due to limited access to compute.

| Model | Input Sequence | [MASK] Tokens |
|---|---|---|
| FULL TOKEN | mars is about [MASK] the [MASK] of earth | half size |
| F | mars is about [MASK] the [MASK] of earth | half size |
| M | mars is about [MASK] the [MASK] of earth | half size |
| L | mars is about [MASK] the [MASK] of earth | half size |
| FL | mars is about [MASK] the [MASK] of earth | half size |
| FF | mars is about [MASK] the [MASK] of earth | half size |
| LL | mars is about [MASK] the [MASK] of earth | half size |
| FML | mars is about [MASK] the [MASK] of earth | half size |
| FFF | mars is about [MASK] the [MASK] of earth | half size |
| LLL | mars is about [MASK] the [MASK] of earth | half size |
| V | mars is about [MASK] the [MASK] of earth | half size |
| C | mars is about [MASK] the [MASK] of earth | half size |

Table 1: Example of a masked input sequence for each model considered. F denotes *first*, M *middle*, L *last*, V *vowels* and C *consonants*.

| Model Category | Vocab Size | #Params |
|---|---|---|
| FULL TOKEN | 50,010 | 124M |
| ONE CHARACTER | 36 | 86M |
| TWO CHARACTERS | 686 | 87M |
| THREE CHARACTERS | 17,586 | 100M |
| VOWELS | 3,690 | 89M |
| CONSONANTS | 50,010 | 124M |

Table 2: Statistics for each model category. We use the most frequent 50K tokens in both FULL TOKEN and CONSONANTS.

1M steps using 8 NVIDIA Tesla V100 (SXM2 - 32GB). We use a learning rate of 1e-4 and a batch size of 16 per GPU. We also use a weight decay of 0.01, attention dropout of 0.1, 10,000 warmup steps, 1e-8 Adam $\epsilon$, 0.9 Adam $\beta_1$ and 0.999 Adam $\beta_2$.

### 4.3 Pre-training Data

The models are pre-trained using the BookCorpus (Zhu et al., 2015) and English Wikipedia datasets from Hugging Face.[4] We convert text to lowercase in both datasets. For the English Wikipedia, we eliminate headers and extract training samples that are no longer than 512 tokens. For the BookCorpus, sentences are concatenated while maintaining a maximum total of 512 tokens. The total number of training samples is 8.1M.

---

[4] https://github.com/huggingface/datasets

### 4.4 Tokenization

In our experiments, we focus on full words. Subword tokenization such as BPE (Sennrich et al., 2016) results in more fragmented inputs with generally shorter input tokens. Therefore, we apply a whitespace tokenizer to split the input sequences into tokens. We replace numbers with a special token <num>, and we filter out some special characters. As a result, we obtain approximately 2.1B tokens from all training samples. Appendix A shows the distribution of token lengths as well as the number of unique tokens for various token lengths in both the pre-training dataset and the downstream tasks training sets.

Table 2 shows the vocabulary size of each model category together with the number of parameters. For instance, the vocabulary of the single character model category consists of the 26 English alphabet letters plus 10 tokens including punctuation marks and special tokens.

### 4.5 Model Fine-tuning

**GLUE** We fine-tune our models using the General Language Understanding Evaluation (GLUE) benchmark (Wang et al., 2018) for up to 20 epochs with early stopping tolerance of 5. We use five different seeds for each fine-tuning task and report the average. We report matched accuracy for MNLI task (Williams et al., 2018), Matthews correlation for CoLA task (Warstadt et al., 2019), Spearman correlation for STS-B task (Cer et al., 2017), accuracy for MRPC task (Dolan and Brockett, 2005),

F1 scores for QQP [5] task, and accuracy for all other tasks. For all models, we retain the same type of input used during pre-training.

**SuperGLUE** We also fine-tune our models on six tasks from the SuperGLUE benchmark (Wang et al., 2019). The tasks are namely Commitment-Bank (CB) (De Marneffe et al., 2019), Choice of Plausible Alternatives (COPA) (Roemmele et al., 2011), Recognizing Textual Entailment (RTE) (Dagan et al., 2010), Words in Context (WiC) (Pilehvar and Camacho-Collados, 2019), The Multi-Sentence Reading Comprehension (MultiRC) (Khashabi et al., 2018) and BoolQ (Clark et al., 2019). Similar to fine-tuning on GLUE, we fine-tune our models for up to 20 epochs with early stopping tolerance of 5. We also use five different seeds for each SuperGLUE task and we report the average score. Similar to GLUE tasks, we retain the original model input type.

### 4.6 Probing Settings

Apart from the downstream predictive performance, we also test whether our character subset models acquire linguistic knowledge by probing them. For this purpose, we employ six widely used probing tasks from Conneau et al. (2018) to analyze the representation output of our various PLMs at each layer. These tasks assess syntactic and semantic information. More details about the probing tasks can be found in Appendix B.

## 5 Results

### 5.1 GLUE

Table 3 presents the results of fine-tuning the models on GLUE. Overall, we observe that using more characters from the tokens results in better performance. For example, the model pre-trained using the last two characters (LL) achieves an average GLUE performance of 68.2%, while the model pre-trained on the last character only (L) achieves an average GLUE performance of 60.1%. Similarly, the performance of the FF model on the MNLI task increases to 71.5% compared 61.4% of the F model. The same can be observed when the number of characters increases from two to three. For instance, in the SST task, the performance of the FFF model increases by 9.3% compared to the performance of the FF model. This boost in performance

is expected as increasing the amount of information available from each token helps models learn better.

The results also show that the models pre-trained on first characters achieve better average performance than the models pre-trained on last characters. For example, the average performance of the F and L models is 62.0% and 60.1%, respectively. Furthermore, the average performance of the FFF model is about 2.8% higher than the average performance of the LLL model. While this demonstrates that the first characters are more informative than the last characters, the results also show that the first and last characters are more informative than the first two or last two characters. For instance, the average performance of the FL model is 2.1% and 2.4% better than the average performance of the FF and LL models, respectively. A similar observation has been made for humans suggesting that the first and last letters of a word are more crucial for human reading comprehension than the middle letters (Grainger et al., 2004; Johnson and Eisler, 2012).

The results also demonstrate that the model pre-trained using only consonants (C) achieves the best performance across all the GLUE tasks compared to the rest of the models trained on partial tokens, only 2.3% less than the average of the full token model. This is most likely because it has suffered less information loss compared to other models as we discuss in §6.2.

Furthermore, we see that the performance retention for the models pre-trained using only one character of the input tokens is high when compared with the performance of the model pre-trained using the full tokens. For example, the F model retains about 77% of the average performance of the model pre-trained using full tokens. Similarly, the model pre-trained using only the middle character achieves approximately 84% of the performance of the model pre-trained using full tokens on the same QNLI task. This might suggest that PLMs have the ability to still learn downstream task-specific associations using very little information from the input tokens.

### 5.2 SuperGLUE

Table 4 presents the fine-tuning results on SuperGLUE. Similar to the fine-tuning results on GLUE, we first observe that the performance of the models increases when more characters from the tokens

---

[5]https://quoradata.quora.com/First-Quora-Dataset-Release-Question-Pairs

| Model | MNLI | QNLI | QQP | RTE | SST | MRPC | CoLA | STS | GLUE Avg. |
|---|---|---|---|---|---|---|---|---|---|
| Token | **82.5** | **89.7** | **86.2** | **67.8** | **91.8** | **86.1** | **58.0** | **86.1** | **81.0 ± 0.3** |
| F | 61.4 | 76.5 | 78.8 | 58.2 | 64.3 | 78.0 | 9.4 | 69.3 | 62.0 ± 0.4 |
| M | 57.5 | 74.8 | 76.9 | 56.7 | 62.3 | 77.6 | 11.4 | 67.3 | 60.6 ± 0.3 |
| L | 57.8 | 74.3 | 77.1 | 58.0 | 64.4 | 75.9 | 12.1 | 61.4 | 60.1 ± 0.3 |
| FL | 71.9 | 83.5 | 84.0 | 58.2 | 77.8 | 83.0 | 27.3 | 79.3 | 70.6 ± 0.4 |
| FF | 71.5 | 83.6 | 83.2 | 57.6 | 76.7 | 83.0 | 11.8 | 80.7 | 68.5 ± 1.4 |
| LL | 68.1 | 81.7 | 82.4 | 58.3 | 75.3 | 80.7 | 25.2 | 74.0 | 68.2 ± 0.2 |
| FML | 78.3 | 87.3 | 85.4 | 60.4 | 87.7 | 82.5 | 47.6 | 83.7 | 76.6 ± 0.2 |
| FFF | 77.9 | 87.8 | 85.2 | 60.3 | 86.0 | 83.3 | 39.3 | 84.6 | 75.5 ± 0.3 |
| LLL | 73.1 | 85.7 | 83.9 | 59.2 | 81.5 | 82.6 | 38.3 | 77.0 | 72.7 ± 0.2 |
| V | 61.8 | 79.9 | 80.5 | 58.3 | 68.2 | 79.4 | 8.7 | 72.1 | 63.6 ± 0.5 |
| C | 80.7 | 88.9 | 85.9 | 61.4 | 90.4 | 84.5 | 52.1 | 85.5 | 78.7 ± 0.4 |

Table 3: Results on GLUE dev sets with standard deviations over five runs for models pre-trained to predict the partial token. **Bold** values denote the best performance across all models. Underlined values denote the second best performance.

| Model | BoolQ | CB | COPA | RTE | WiC | MultiRC | SuperGLUE Avg. |
|---|---|---|---|---|---|---|---|
| Majority | 62.2 | 50.0 | 55.0 | 52.7 | 50.0 | 59.9 | 55.0 |
| Token | **73.5** | **83.5** | **61.6** | **66.2** | **66.5** | **69.7** | **70.2 ± 0.8** |
| F | 66.6 | 73.7 | 57.9 | 56.6 | 59.3 | 65.3 | 63.2 ± 1.1 |
| M | 66.6 | 70.3 | 57.2 | 57.9 | 57.4 | 64.7 | 62.4 ± 0.9 |
| L | 66.3 | 68.8 | 56.8 | 56.4 | 57.2 | 65.0 | 61.8 ± 0.8 |
| FL | 70.1 | 80.8 | 59.5 | 58.7 | 59.3 | 68.2 | 66.1 ± 0.6 |
| FF | 69.8 | 80.1 | 58.1 | 57.9 | 60.7 | 68.3 | 65.8 ± 0.9 |
| LL | 69.4 | 73.1 | 57.6 | 57.6 | 58.6 | 66.3 | 63.8 ± 1.1 |
| FML | 72.4 | 79.5 | 60.2 | 59.5 | 63.7 | 69.1 | 67.4 ± 0.7 |
| FFF | 70.8 | 80.4 | 59.4 | 59.8 | 61.9 | 68.8 | 66.9 ± 0.9 |
| LLL | 70.1 | 75.9 | 58.8 | 57.9 | 60.1 | 67.8 | 65.1 ± 0.9 |
| V | 68.4 | 68.3 | 57.5 | 57.7 | 59.8 | 66.8 | 63.1 ± 0.8 |
| C | 72.1 | 82.9 | 60.0 | 60.8 | 62.5 | 68.0 | 67.7 ± 0.7 |

Table 4: Results on six SuperGLUE task dev sets with standard deviations over five runs. **Bold** values denote the best performance across all models. Underlined values denote the second best performance.

are used in pre-training. For instance, the average performance of the model increases from 63.2%, when pre-trained using only the first character, to 65.8% when pre-trained using the first two characters. A similar pattern can be observed when pre-training the model using the last three characters, we notice an increase of 1.3% of the average performance compared with the model pre-trained using only the last two characters when predicting the character parts of the token. That indicates that PLMs learn better when the information from each token increases.

The results also show consistent patterns to the results on GLUE where the first characters are more informative than the last characters. For example, the FFF model achieves an average performance

of 66.9% while the LLL model 65.1%. Similarly, the model pre-trained using the first character (F) achieves an accuracy of 73.7% while the model pre-trained using the last three characters (L) achieves an accuracy of 68.8% on the CB task.

We also observe, in a similar pattern to the fine-tuning results on GLUE, that the model pre-trained using only consonant characters (C) achieves the second-best average performance of 67.7%. The results also show that the C model achieves 4.6% higher average performance compared to the model pre-trained using only vowel characters (V). A comparable observation has been made in the context of human reading comprehension, indicating that consonant characters hold greater significance compared to vowel characters (Fry, 1968).

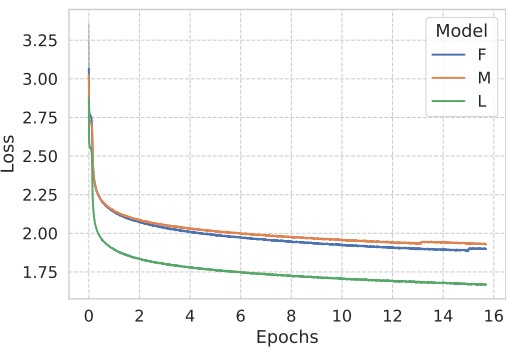

Figure 2: The loss curves for single character models (F, M and L).

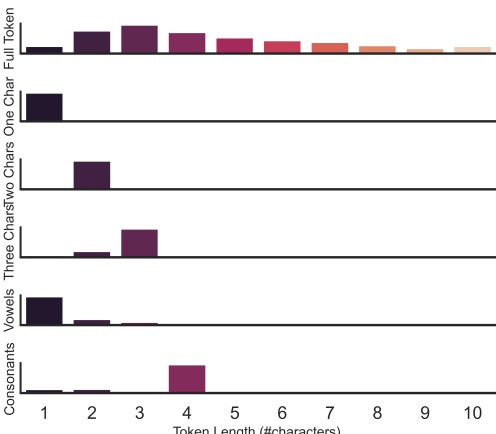

Figure 3: Token length distribution of the pre-training corpus for each model category. 100% of tokens for ONE CHAR have one letter, 99% for TWO CHARS have two and 79% for THREE CHARS have three. For VOWELS 73% are single-character tokens and for CONSONANTS 78% of pre-training tokens consist of four characters.

Finally, the results also demonstrate substantial performance retention for the models pre-trained using subsets of each input token compared with the performance of the model pre-trained using full input tokens. For instance, the model pre-trained using only the first character retains approximately 90% of the average performance of the model pre-trained using full tokens. In addition, the LL model achieves retention in the performance of approximately 94% on the BoolQ task compared with the model pre-trained using the full tokens. However, this might be attributed to the fact that SuperGLUE is a more difficult benchmark where the performance of the full token model is not as high as in GLUE (average 81.0% and 70.2% respectively).

## 6 Analysis

### 6.1 Pre-training Loss Curves

Figure 2 shows the pre-training loss curves for the single character models illustrating the progression and convergence of the loss function throughout the pre-training process. We observe that both the F and M models have very close pre-training losses and that the model L has the lowest pre-training loss. Overall, we note that the three models have stable pre-training which indicates that the models are able to learn using limited input information. The pre-training loss curves for the other pre-trained models can be found in Appendix C.

### 6.2 Corpus-based Token Information Loss

We aim to analyze how much information loss occurs when we use different tokenization strategis for each of the models considered (§3). We combine all models in Table 1 in three categories depending on how many characters they keep in each word during tokenization: ONE CHAR, TWO

CHARS and THREE CHARS respectively. In Figure 3 we plot the distribution of token lengths (in terms of number of characters) in each pre-training corpus after tokenization.[6] We observe that the distribution of the FULL TOKEN pre-training corpus is more evenly distributed, with 42% of the total pre-training tokens to include either one, two or three characters. When we apply our tokenizers for ONE, TWO and THREE CHARS model categories of models, we remove all the extra characters of the words to keep the threshold of the required character length. For instance, for the THREE CHARS every token longer than three letters is trimmed to exact three letters. This way, we can use this corpus-based metric to measure how much information is lost from the original corpus (via its vocabulary). For ONE CHAR 95% of the original words are trimmed down to a single letter, while in TWO CHARS 79% are turned into double-character tokens. For the THREE CHARS category, we lose around 48% of the original tokens with more than 3 letters, while the resulting distribution is 5% single, 16% double and 79% triple-character tokens.

We also measure the distribution of n-grams between the different model categories. Table 6 shows the number of unigrams, bigrams and trigrams in the pre-training corpus. We observe that the FULL TOKEN and the CONSONANTS models

---

[6]We plot the distribution of all the tokens in the pre-training corpora, not the unique tokens.

| Model | TreeDepth (Syntactic) | TopConst (Syntactic) | BShift (Syntactic) | Tense (Semantic) | SubjNum (Semantic) | CoordInv (Semantic) |
|---|---|---|---|---|---|---|
| Majority | 17.9 | 5.0 | 50.0 | 50.0 | 50.0 | 50.0 |
| Full-Token | 42.7 ± 0.1 | 78.4 ± 0.3 | **88.5 ± 0.3** | **89.2 ± 0.8** | 88.0 ± 0.4 | **73.2 ± 0.3** |
| F | 31.3 ± 0.3 | 53.3 ± 0.3 | 56.0 ± 0.4 | 63.4 ± 0.9 | 61.2 ± 0.2 | 59.6 ± 0.3 |
| M | 30.6 ± 0.2 | 53.7 ± 0.2 | 55.2 ± 0.2 | 63.3 ± 0.3 | 65.9 ± 0.2 | 58.4 ± 0.4 |
| L | 33.2 ± 0.2 | 60.4 ± 0.3 | 57.3 ± 0.3 | 84.1 ± 0.5 | 83.6 ± 0.0 | 60.4 ± 0.5 |
| FL | 43.3 ± 0.6 | 78.6 ± 0.4 | 72.0 ± 0.8 | 85.3 ± 0.3 | 88.1 ± 0.7 | 68.2 ± 0.3 |
| FF | 39.9 ± 0.6 | 69.0 ± 0.6 | 67.0 ± 0.4 | 65.2 ± 0.4 | 72.0 ± 0.5 | 63.5 ± 0.4 |
| LL | 41.0 ± 0.5 | 74.9 ± 0.4 | 69.5 ± 0.3 | 88.0 ± 0.2 | 89.2 ± 0.2 | 64.5 ± 0.3 |
| FML | **45.0 ± 0.4** | **81.4 ± 0.3** | 84.5 ± 0.2 | 87.0 ± 0.3 | **91.3 ± 0.2** | 68.4 ± 0.8 |
| FFF | 41.6 ± 1.0 | 79.3 ± 0.3 | 80.2 ± 0.4 | 68.2 ± 0.1 | 77.6 ± 0.2 | 67.7 ± 1.2 |
| LLL | 44.8 ± 0.2 | 81.1 ± 0.3 | 82.0 ± 0.2 | 88.4 ± 0.1 | 90.0 ± 0.2 | 69.1 ± 0.5 |
| V | 30.4 ± 0.4 | 51.4 ± 0.4 | 57.6 ± 0.3 | 68.3 ± 0.1 | 65.6 ± 0.1 | 57.9 ± 0.2 |
| C | 44.2 ± 0.6 | 80.2 ± 0.1 | 85.0 ± 0.2 | 87.1 ± 0.4 | 78.9 ± 0.3 | 71.0 ± 0.2 |

Table 5: Results on the probing tasks for the best-performing layer of each model using mean accuracy with standard deviation over three runs. **Bold** values denote the best performance across each task. Underlined values denote the second best performance.

| Category | Unigrams | Bigrams | Trigrams |
|---|---|---|---|
| FULL TOKEN | 50K | 43M | 302M |
| ONE CHAR | 34 | 1K | 37K |
| TWO CHARS | 686 | 300K | 20M |
| THREE CHARS | 17K | 13M | 185M |
| VOWELS | 3K | 2M | 28M |
| CONSONANTS | 50K | 39M | 299M |

Table 6: The number of unigrams, bigrams and trigrams in the pre-training corpus for each model category.

have the largest number of unigrams, bigrams and trigrams compared to other model categories. We also see that the tokenized pre-training corpus of the TWO CHARS model retains only about 7% of the total number of trigrams in the pre-traing corpus of the FULL TOKEN. For the THREE CHARS category, we observe a reduction of 66% and 70% in the number of unigrams and bigrams, respectively, compared to the FULL TOKEN. This indicates a substantial decrease in the occurrence of unique unigrams and bigrams when utilizing only three characters as opposed to full tokens.

Overall our analysis shows that even though we lose a very large percentage of the original information (in terms of token characters) from the original pre-training corpus for the full token model, surprisingly, fine-tuning the models pre-trained on partial tokens in downstream tasks does not hurt performance in an analogous way (§5).

## 6.3 Probing

We hypothesize that the probing performance of models pre-trained on partial tokens will be lower to the performance of full tokens. Table 5 presents the results on the six probing tasks using the representations from the models as inputs to the MLP classifier. Note that we evaluate all layers from each model individually and choose the best-performing layer on the test set.

Overall, we find that the predictive performance of the model trained on representations learned using the first, middle and last (FML) characters of input tokens achieves the best performance in three out of the six probing tasks. For example, in the SubjNum probing task, the FML model achieves the best performance of 91.3%, while the one using only the first character of input tokens achieves 61.2%. This is surprising given the large information loss in the pre-training data (§6.2).

We also see that the ability of PLMs to acquire linguistic information increases when more characters from the tokens are used in pre-training. For instance, in the TopConst probing task, the model pre-trained using the first character achieves an accuracy of 53.3%, while the model pre-trained using the first two characters achieves an accuracy of 69.0%. Similarly, in the CoordInv probing task, the model pre-trained using the last three characters achieves 4.6% higher accuracy compared with the model pre-trained using the last two characters.

The results also show that models pre-trained

using the last characters perform better than the first character models. For example, in the Tense probing task, the L model achieves an accuracy of 84.1%, while the F model achieves an accuracy of 63.4%. In the BShift task, the LLL model achieves 1.8% higher accuracy than the FFF model. This might suggest that specific character positions within tokens play a crucial role in how PLMs encode linguistic information in English.

Finally, the results show that even when models are pre-trained using one character, they are still able to encode some linguistic information. For instance, in the Tense probing task, the model pre-trained using only the last character achieves an accuracy of 84.1% while the the model pre-trained using full tokens achieves 89.2%. In the SubjNum task, the difference in accuracy between the L and full-token models is only 4.4%. This indicates similar behavior to downstream tasks in GLUE and SuperGLUE. The MLPs used for modeling the probing tasks are able to find associations in the data using little information from the token.

## 7 Discussion

Our results reveal that certain subsets of input tokens are more informative than others. For instance, we found that PLMs perform better when pre-trained using the first and last characters instead of the first or last two characters of input tokens. This appears to align with research in cognitive science and psycholinguistics which suggests that the first and last letters of a word are more crucial for human reading comprehension than the interior letters and that the positions of letters are not given equal weight in the cognitive representation of a word (Grainger et al., 2004; Johnson and Eisler, 2012). Previous studies have also suggested that the first and last letters of a word hold more importance because they pertain to the way the mind organizes and retrieves lexical information (Forster, 1976; Jordan, 1990; Jordan et al., 2003). Other studies have suggested that the first and last letters of a word are more easily identified as they are less impacted by external interference or the presence of surrounding letters compared to inner letters (Bouma, 1973; Chambers, 1979; McCusker et al., 1981; van der Heijden, 2003).

Surprisingly, our results also highlight that PLMs are still able to learn from data that may be incomprehensible to humans. Our models perform well on natural language understanding tasks or appear to 'learn' semantic and syntactic information (i.e. probing) by using only one or two characters from each token. This demonstrates the ability of PLMs to find patterns in seemingly random or limited data. Additionally, our findings also suggest that these models may have the capability to process language in ways that are fundamentally different from human cognition.

## 8 Conclusion

In this work, we explored how the pre-training of language models using different subsets of characters from the input tokens affect their performance on downstream tasks. To our surprise, we found that even using just one character from each input token achieves performance way above chance. Our probing results further revealed that specific character positions within tokens affect how the models capture linguistic knowledge.

## Acknowledgments

AA is supported by the Centre for Doctoral Training in Speech and Language Technologies (SLT) and their Applications funded by UK Research and Innovation grant EP/S023062/1. NA is supported by EPSRC grant EP/V055712/1, part of the European Commission CHIST-ERA programme, call 2019 XAI: Explainable Machine Learning-based Artificial Intelligence.

## Limitations

**Languages** Our research is currently limited only to English due to computational constraints. However, expanding to other languages with different characteristics presents a potential area for future exploration.

**Models** We employed a BERT-like model (Devlin et al., 2019) for simplicity and due to limited access to compute. Exploring different model sizes or types, i.e. generative models such as GPT-3 (Brown et al., 2020) could be considered in future studies.

**Model Usability** We experimented with models trained on partial tokens, even only on single characters. We do not claim that these models have any practical usability in downstream tasks apart from helping us answering our research questions. Our goal in this study was to test the boundaries of PLMs by pre-training on data containing very limited information.

**Probing Tasks** We experiment with six probing tasks from Conneau et al. (2018) as a representative sample. Expanding probing tasks to cover more linguistic characteristics is another avenue for future research.

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

# Appendix

## A   Distribution of Token Lengths

### A.1   Pre-training Dataset

Figures 4 show the distribution of the token lengths in the pre-training dataset.

### A.2   GLUE Tasks

Figures 5 to 12 show the distribution of the token lengths in the training set of each GLUE task.

### A.3   SuperGLUE Tasks

Figures 13 to 16 show the distribution of the token lengths in the training set of each SuperGLUE task.

## B   Probing Tasks

The syntactic information tasks include: *TreeDepth*, which examines if representations maintain information about the hierarchical structure of a sentence by predicting the depth of its parse tree. *TopConst*, which predicts the top constituents of a sentence's parse tree. *BShift* evaluates if two neighboring words have been reversed or not. The semantic information tasks include *Tense*, which predicts if the verb in the main clause is in present or past form. *SubjNum* predicts whether the subject of the main clause is singular or plural. *CoordInv* determines if a sentence consisting of two coordinate clauses has been inverted or not. Each task consists of 100k sentences for training, 10k sentences for validation and 10k sentences for testing. Using the recommended hyperparameters in the SentEval toolkit (Conneau and Kiela, 2018), we train a multi-layer perceptron (MLP) classifier for each task.

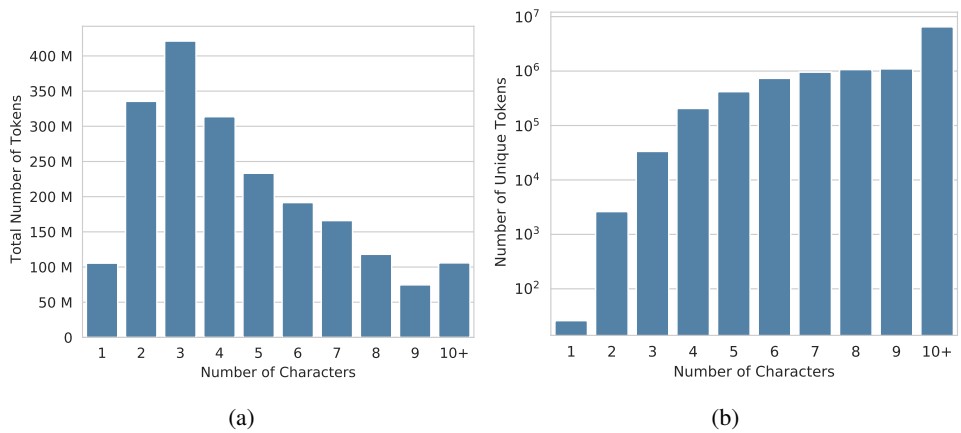

(a)        (b)

Figure 4: The distribution of the token lengths in the pre-training dataset.

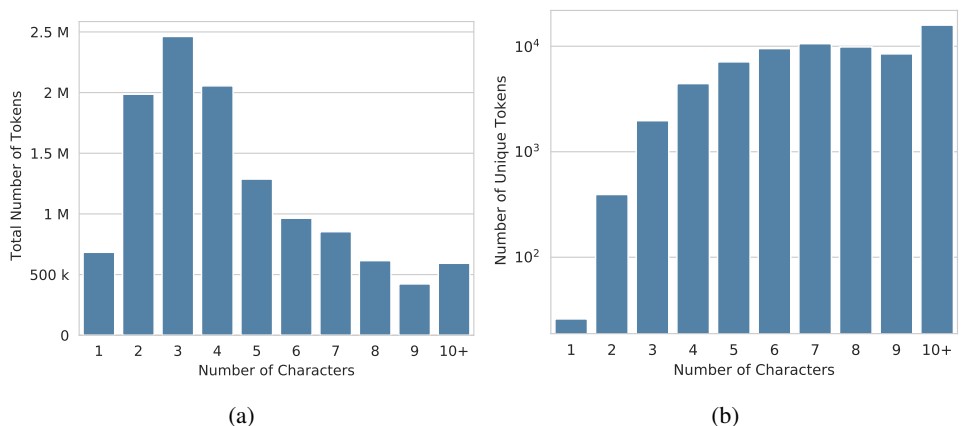

(a)        (b)

Figure 5: The distribution of the token lengths in the training set of the MNLI GLUE task.

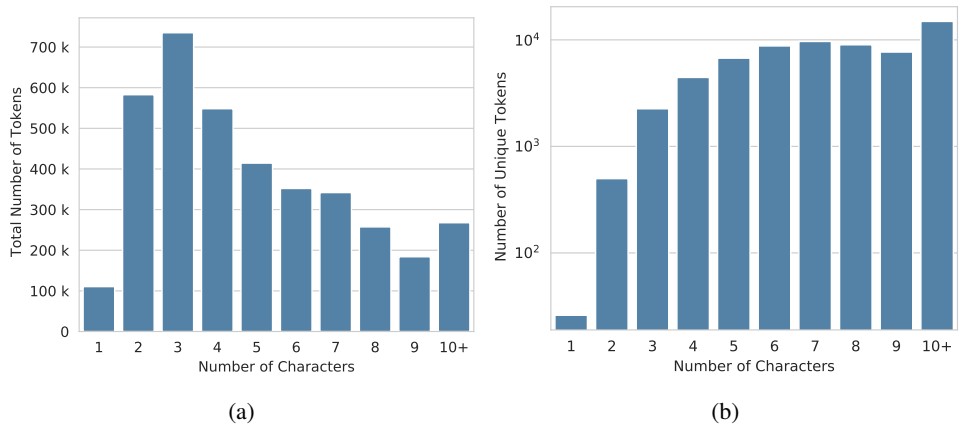

(a)        (b)

Figure 6: The distribution of the token lengths in the training set of the QNLI GLUE task.

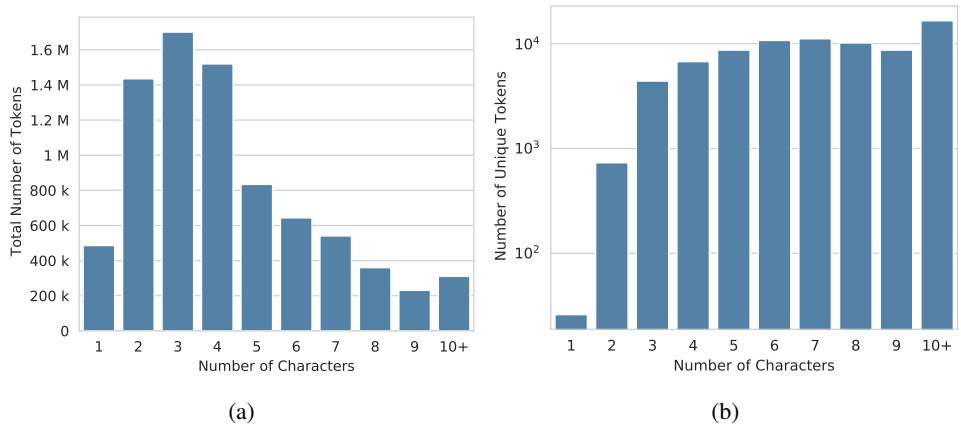

Figure 7: The distribution of the token lengths in the training set of the QQP GLUE task.

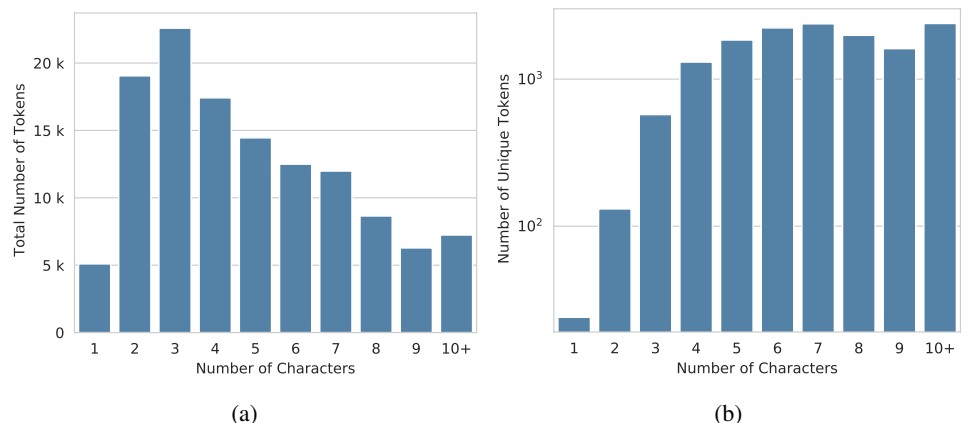

Figure 8: The distribution of the token lengths in the training set of the RTE GLUE task.

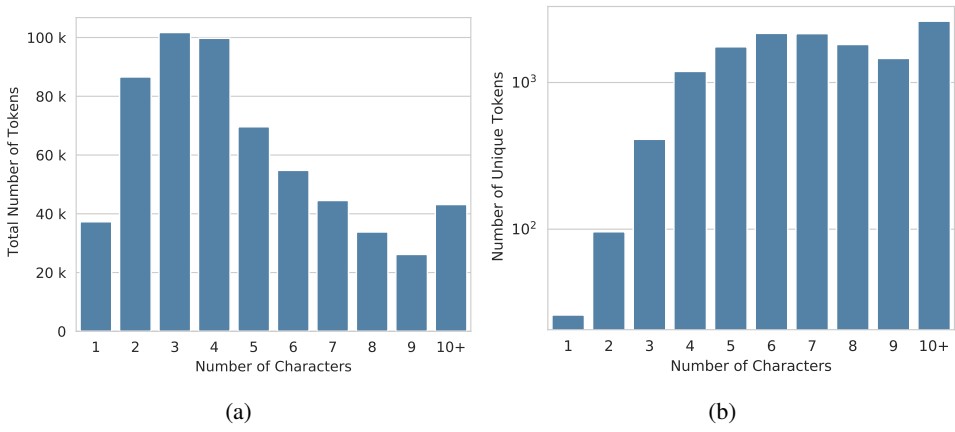

Figure 9: The distribution of the token lengths in the training set of the SST GLUE task.

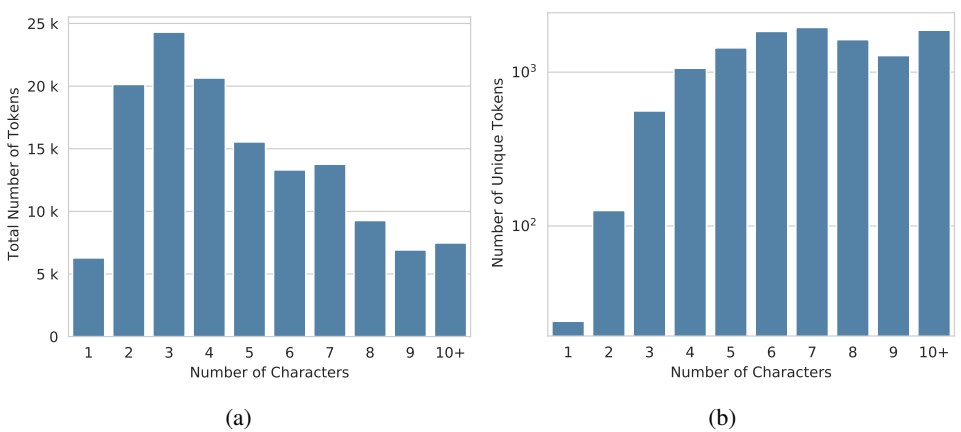

(a)                                           (b)

Figure 10: The distribution of the token lengths in the training set of the MRPC GLUE task.

## C  Pre-training Loss

### C.1  Two Characters

Figure 17 shows the pre-training loss curves for the models pre-trained using two characters from each input token. Similar to the loss curves of single character models, we observe that the two characters models have stable pre-training.

### C.2  Three Characters

Figure 18 shows the pre-training loss curves for the models pre-trained using three characters from each input token. We also note that the models have stable pre-training.

### C.3  Token, Vowels and Consonants

Figure 19 shows the pre-training loss curves for the model pre-trained using full tokens in addition to the models pre-trained using the vowel and consonant characters from each input token.

## D  GLUE Full Results

Table 7 show the results for all models pre-trained using both pre-training tasks, predicting the character parts of the token and predicting the original full token. The results show that the pre-training task of predicting the character parts of the token slightly helps most models achieve better performance than the other pre-training task of predicting the original full token. For instance, the FML model achieves an average performance of 76.6% when pre-trained to predict the character parts of the token, while it achieves an average performance of 76.3% when pre-trained to predict the original full token. Similarly, the FL model achieves about 0.3% higher average performance when pre-trained to predict the character parts of the token compared to when pre-trained to predict the original token. This might indicate that the pre-training task to predict the original full token using subsets of the token is a hard task for models as the size of the full tokens vocabulary (used in the output classification layer of the model) is much higher than the size of the characters subsets vocabulary (used in the input).

## E  SuperGLUE Full Results

Table 8 show the results for all models pre-trained using both pre-training tasks, predicting the character parts of the token and predicting the original full token, on the six SuperGLUE tasks. Similar to GLUE, the the results show that the models do not benefit much when they are pre-trained to predict the full token instead of predicting the character parts of the token. For instance, the M model achieves an average accuracy of 62.4% when pre-trained to predict the character parts of the token while it achieves an average accuracy of 62.1% when pre-trained to predict the full token. The same can be observed in the performance of the FL models on the MultiRC task, where the difference in performance between the two pre-training tasks is 0.5%.

## F  Fine-tuning the Full Token Model

Tables 9 and 10 show the results of fine-tuning the full token model (Token) using partial input token on both GLUE and SuperGLUE benchmarks, respectively. The results show the full token model achieves lower performance when fine-tuned using partial tokens on both GLUE and SuperGLUE benchmarks compared to the models pre-trained and fine-tuned using the same partial input token.

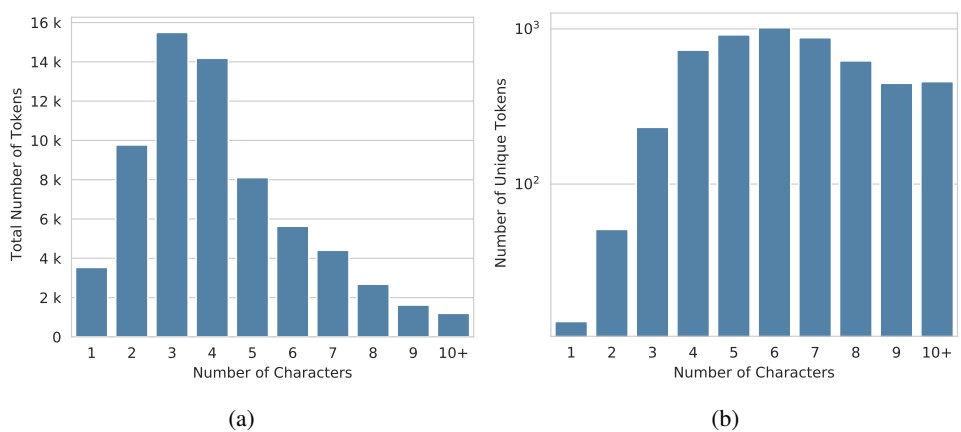

(a)            (b)

Figure 11: The distribution of the token lengths in the training set of the CoLA GLUE task.

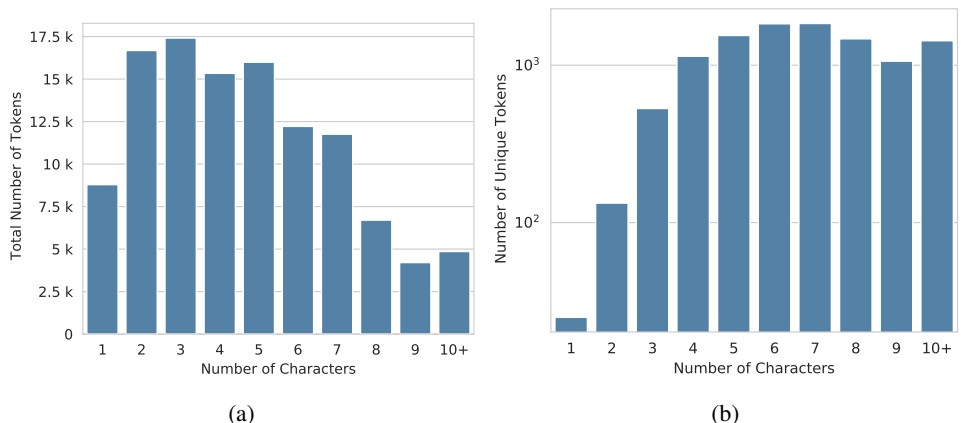

(a)            (b)

Figure 12: The distribution of the token lengths in the training set of the STS GLUE task.

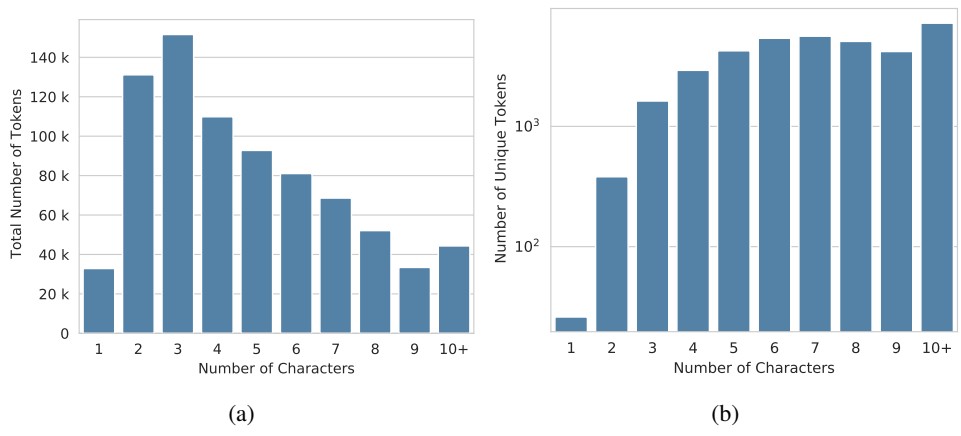

(a)            (b)

Figure 13: The distribution of the token lengths in the training set of the BoolQ SuperGLUE task.

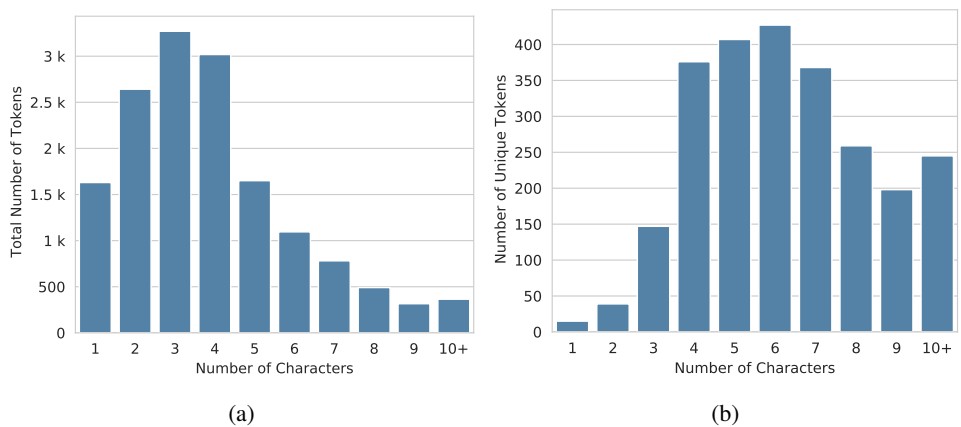

Figure 14: The distribution of the token lengths in the training set of the CB SuperGLUE task.

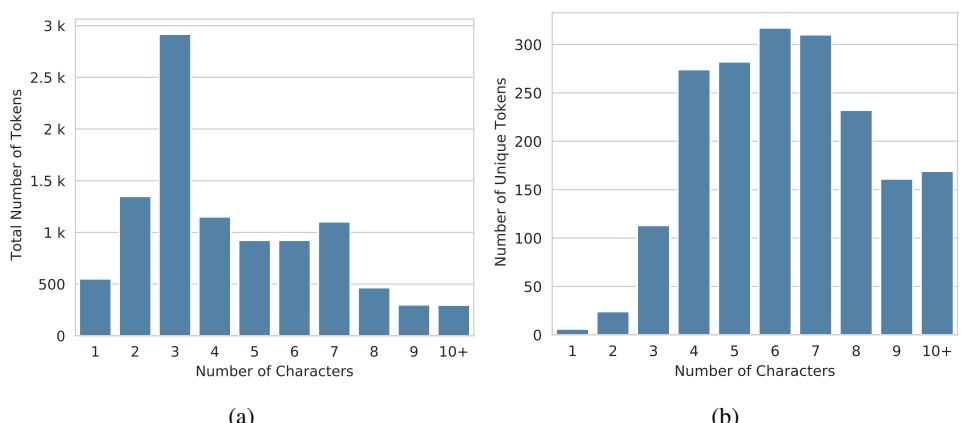

Figure 15: The distribution of the token lengths in the training set of the COPA SuperGLUE task.

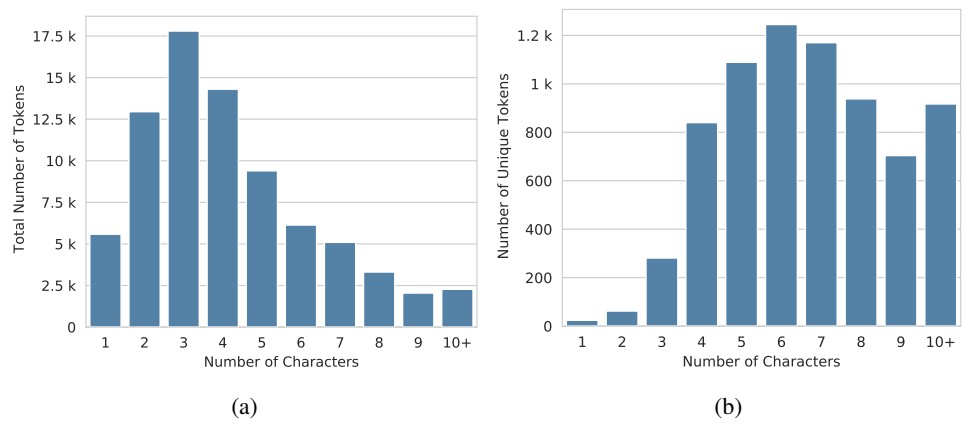

Figure 16: The distribution of the token lengths in the training set of the WiC SuperGLUE task.

| Model | MNLI | QNLI | QQP | RTE | SST | MRPC | CoLA | STS | GLUE Avg. |
|---|---|---|---|---|---|---|---|---|---|
| Token | **82.5** | **89.7** | **86.2** | **67.8** | **91.8** | **86.1** | **58.0** | **86.1** | **81.0 ± 0.3** |
| | | | | Pre-training: Predicting the partial token | | | | | |
| F | 61.4 | 76.5 | 78.8 | 58.2 | 64.3 | 78.0 | 9.4 | 69.3 | 62.0 ± 0.4 |
| M | 57.5 | 74.8 | 76.9 | 56.7 | 62.3 | 77.6 | 11.4 | 67.3 | 60.6 ± 0.3 |
| L | 57.8 | 74.3 | 77.1 | 58.0 | 64.4 | 75.9 | 12.1 | 61.4 | 60.1 ± 0.3 |
| FL | 71.9 | 83.5 | 84.0 | 58.2 | 77.8 | 83.0 | 27.3 | 79.3 | 70.6 ± 0.4 |
| FF | 71.5 | 83.6 | 83.2 | 57.6 | 76.7 | 83.0 | 11.8 | 80.7 | 68.5 ± 1.4 |
| LL | 68.1 | 81.7 | 82.4 | 58.3 | 75.3 | 80.7 | 25.2 | 74.0 | 68.2 ± 0.2 |
| FML | 78.3 | 87.3 | 85.4 | 60.4 | 87.7 | 82.5 | 47.6 | 83.7 | 76.6 ± 0.2 |
| FFF | 77.9 | 87.8 | 85.2 | 60.3 | 86.0 | 83.3 | 39.3 | 84.6 | 75.5 ± 0.3 |
| LLL | 73.1 | 85.7 | 83.9 | 59.2 | 81.5 | 82.6 | 38.3 | 77.0 | 72.7 ± 0.2 |
| V | 61.8 | 79.9 | 80.5 | 58.3 | 68.2 | 79.4 | 8.7 | 72.1 | 63.6 ± 0.5 |
| C | 80.7 | 88.9 | 85.9 | 61.4 | 90.4 | 84.5 | 52.1 | 85.5 | 78.7 ± 0.4 |
| | | | | Pre-training: Predicting the full token | | | | | |
| F | 61.6 | 76.3 | 78.7 | 55.2 | 66.2 | 75.9 | 11.7 | 69.5 | 61.9 ± 0.3 |
| M | 57.2 | 75.1 | 77.4 | 55.6 | 63.5 | 75.4 | 4.7 | 67.6 | 59.6 ± 0.6 |
| L | 57.4 | 74.1 | 77.6 | 54.7 | 65.3 | 74.5 | 10.5 | 62.3 | 59.5 ± 0.7 |
| FL | 72.6 | 84.1 | 84.3 | 54.6 | 78.6 | 80.7 | 29.8 | 77.9 | 70.3 ± 0.4 |
| FF | 71.4 | 83.6 | 83.3 | 56.5 | 77.8 | 83.3 | 19.1 | 81.2 | 69.5 ± 0.2 |
| LL | 68.3 | 81.6 | 82.3 | 56.6 | 76.8 | 81.0 | 24.1 | 73.6 | 68.1 ± 0.5 |
| FML | 78.0 | 87.0 | 85.3 | 59.7 | 88.3 | 82.8 | 46.9 | 82.7 | 76.3 ± 0.3 |
| FFF | 78.0 | 87.4 | 85.4 | 57.8 | 86.2 | 82.7 | 38.8 | 83.4 | 75.0 ± 0.2 |
| LLL | 73.7 | 85.7 | 84.3 | 57.5 | 82.6 | 81.8 | 38.9 | 76.9 | 72.7 ± 0.4 |
| V | 62.2 | 80.0 | 80.6 | 58 | 69.2 | 80.0 | 9.6 | 71.4 | 63.9 ± 0.7 |
| C | 80.6 | 88.5 | 85.7 | 62.2 | 90.9 | 85.5 | 51.5 | 85.6 | 78.8 ± 0.2 |

Table 7: Full results on GLUE dev sets with standard deviations over five runs. **Bold** values denote the best performance across all models. Underlined values denote the second best performance.

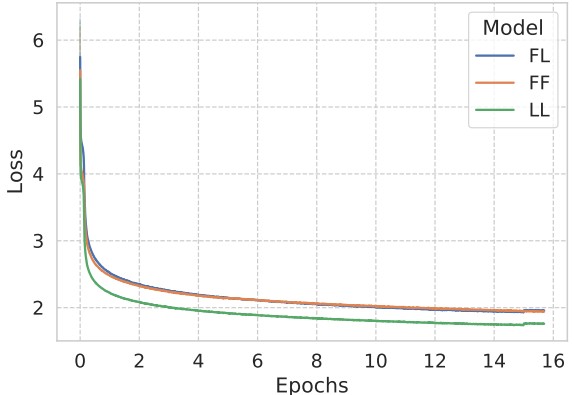

Figure 17: The loss curves for two characters models.

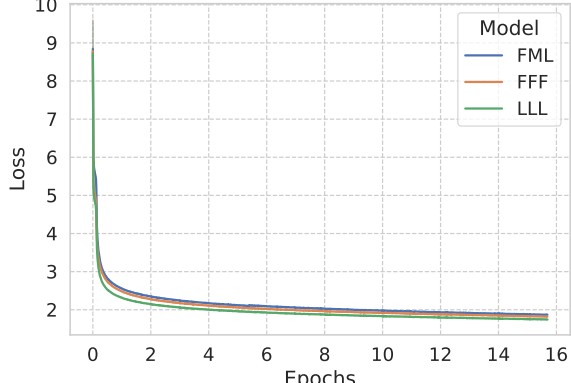

Figure 18: The loss curves for three character models.

| Model | BoolQ | CB | COPA | RTE | WiC | MultiRC | Avg. |
|---|---|---|---|---|---|---|---|
| Majority | 62.2 | 50.0 | 55.0 | 52.7 | 50.0 | 59.9 | 55.0 |
| Token | **73.5** | **83.5** | **61.6** | **66.2** | **66.5** | **69.7** | **70.2 ± 0.8** |
| *Pre-training: Predicting the partial token* | | | | | | | |
| F | 66.6 | 73.7 | 57.9 | 56.6 | 59.3 | 65.3 | 63.2 ± 1.1 |
| M | 66.6 | 70.3 | 57.2 | 57.9 | 57.4 | 64.7 | 62.4 ± 0.9 |
| L | 66.3 | 68.8 | 56.8 | 56.4 | 57.2 | 65.0 | 61.8 ± 0.8 |
| FL | 70.1 | 80.8 | 59.5 | 58.7 | 59.3 | 68.2 | 66.1 ± 0.6 |
| FF | 69.8 | 80.1 | 58.1 | 57.9 | 60.7 | 68.3 | 65.8 ± 0.9 |
| LL | 69.4 | 73.1 | 57.6 | 57.6 | 58.6 | 66.3 | 63.8 ± 1.1 |
| FML | 72.4 | 79.5 | 60.2 | 59.5 | 63.7 | 69.1 | 67.4 ± 0.7 |
| FFF | 70.8 | 80.4 | 59.4 | 59.8 | 61.9 | 68.8 | 66.9 ± 0.9 |
| LLL | 70.1 | 75.9 | 58.8 | 57.9 | 60.1 | 67.8 | 65.1 ± 0.9 |
| V | 68.4 | 68.3 | 57.5 | 57.7 | 59.8 | 66.8 | 63.1 ± 0.8 |
| C | 72.1 | 82.9 | 60.0 | 60.8 | 62.5 | 68.0 | 67.7 ± 0.7 |
| *Pre-training: Predicting the full token* | | | | | | | |
| F | 67.8 | 72.1 | 58.3 | 55.4 | 58.6 | 65.8 | 63.0 ± 0.6 |
| M | 67.6 | 71.7 | 55.2 | 56.1 | 57.0 | 65.0 | 62.1 ± 0.8 |
| L | 67.1 | 71.2 | 58.1 | 55.0 | 57.9 | 65.1 | 62.4 ± 0.9 |
| FL | 69.4 | 70.5 | 59.7 | 57.1 | 60.1 | 67.7 | 64.1 ± 0.9 |
| FF | 69.5 | 72.3 | 58.9 | 56.4 | 59.2 | 68.0 | 64.1 ± 0.8 |
| LL | 68.4 | 70.6 | 58.5 | 56.7 | 58.7 | 67.1 | 63.3 ± 1.0 |
| FML | 72.3 | 79.1 | 62.8 | 59.6 | 61.1 | 68.2 | 67.2 ± 0.8 |
| FFF | 71.6 | 78.4 | 60.8 | 59.8 | 60.9 | 69.3 | 66.8 ± 0.9 |
| LLL | 69.8 | 71.9 | 60.1 | 58.4 | 60.3 | 67.6 | 64.7 ± 1.1 |
| V | 68.2 | 70.6 | 60.8 | 57.3 | 59.1 | 65.9 | 63.7 ± 0.7 |
| C | 72.6 | 80.4 | 59.7 | 61.0 | 64.1 | 69.3 | 67.9 ± 0.9 |

Table 8: Full results on six SuperGLUE task dev sets with standard deviations over five runs. **Bold** values denote the best performance across all models. Underlined values denote the second best performance.

| Partial Token | MNLI | QNLI | QQP | RTE | SST | MRPC | CoLA | STS | GLUE Avg. |
|---|---|---|---|---|---|---|---|---|---|
| F | 59.3 | 74.3 | 77.8 | 56.2 | 64.4 | 76.5 | 7.3 | 66.4 | 60.3 ± 0.4 |
| M | 55.3 | 71.8 | 75.0 | 55.5 | 60.5 | 74.6 | 8.1 | 65.0 | 58.2 ± 0.9 |
| L | 55.7 | 72.3 | 75.9 | 52.3 | 60.1 | 74.8 | 10.3 | 58.8 | 57.5 ± 0.8 |
| FL | 66.2 | 78.3 | 81.4 | 59.4 | 71.2 | 77.6 | **16.1** | 73.9 | 65.5 ± 0.6 |
| FF | 66.3 | 79.1 | 81.3 | 59.4 | 70.3 | 79.5 | 12.0 | 77.9 | 65.7 ± 0.4 |
| LL | 63.0 | 77.3 | 80.3 | 58.6 | 68.7 | 79.7 | 10.6 | 69.8 | 63.5 ± 1.0 |
| FML | 69.5 | 80.1 | 82.4 | 59.7 | 80.3 | 79.1 | 12.9 | 77.6 | 67.7 ± 0.4 |
| FFF | **70.7** | **80.9** | **83.2** | 59.9 | 77.7 | **80.3** | 11.9 | **81.4** | **68.2 ± 0.6** |
| LLL | 67.0 | 79.6 | 81.6 | **60.4** | 73.6 | 79.4 | 13.7 | 73.5 | 66.1 ± 0.4 |
| V | 58.1 | 77.8 | 79.1 | 56.6 | 64.3 | 77.1 | 6.5 | 69.5 | 61.1 ± 0.5 |
| C | 69.1 | 80.2 | 82.3 | 59.7 | **79.9** | 79.5 | 15.4 | 78.1 | 68.0 ± 0.2 |

Table 9: Results for fine-tuning the **Full Token** model on GLUE dev sets using partial input tokens. **Bold** values denote the best performance across all models. Underlined values denote the second best performance.

| Partial Token | BoolQ | CB | COPA | RTE | WiC | MultiRC | Avg. |
|---|---|---|---|---|---|---|---|
| Majority | 62.2 | 50.0 | 55.0 | 52.7 | 50.0 | 59.9 | 55.0 |
| F | 66.0 | 68.8 | 52.0 | 54.1 | 55.6 | 60.4 | 59.5 ± 1.1 |
| M | 66.4 | 64.7 | 57.0 | 56.6 | 55.8 | 61.1 | 60.3 ± 0.9 |
| L | 66.0 | 71.9 | 56.4 | 55.0 | 57.7 | 62.3 | 61.6 ± 0.8 |
| FL | 67.5 | 73.2 | 55.5 | 59.7 | 56.6 | 64.0 | 62.8 ± 1.2 |
| FF | 67.5 | 70.5 | 55.2 | 60.7 | 58.4 | 63.7 | 62.7 ± 1.0 |
| LL | 67.4 | 73.2 | 61.5 | 57.9 | 59.6 | 61.0 | 63.4 ± 0.8 |
| FML | **68.4** | **76.8** | **62.8** | 60.6 | 60.5 | 66.4 | **65.9 ± 0.9** |
| FFF | 67.2 | 69.2 | 56.0 | 59.7 | **60.9** | 64.4 | 62.9 ± 1.1 |
| LLL | 67.1 | 75.4 | 55.0 | **62.6** | 60.3 | **64.7** | 64.2 ± 1.2 |
| V | 66.4 | 69.2 | 56.5 | 56.2 | 57.1 | 63.2 | 61.4 ± 1.1 |
| C | 67.2 | 73.7 | 54.2 | 60.6 | 60.3 | 64.1 | 63.4 ± 0.7 |

Table 10: Results for fine-tuning the **Full Token** model on six SuperGLUE task dev sets using partial input tokens. **Bold** values denote the best performance across all models.

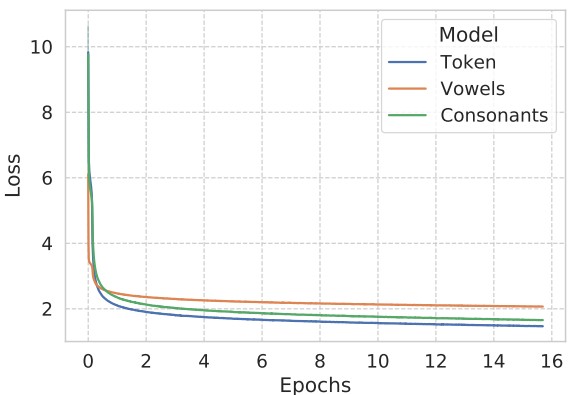

Figure 19: The loss curves for Full Token, Consonants and Vowels models.