# OpenReview forum: "Understanding the Role of Input Token Characters in Language Models: How Does Information Loss Affect Performance?"
_EMNLP/2023/Conference — EMNLP 2023 Main_

### Official Review · Reviewer_Ed3M · 2023-07-30

**Soundness:** 3

**Excitement:**

4: Strong: This paper deepens the understanding of some phenomenon or lowers the barriers to an existing research direction.

**Missing References:**

Each dataset in GLUE and SuperGLUE should be cited in Section 4.5.

**Paper Topic And Main Contributions:**

Topic
===
This paper studies if pre-trained language models (PLM) can work even if characters in the tokens are dropped. They pre-train different PLMs by keeping different numbers of characters (one, two, or three) and different positions (first, last, or middle) in the input token. After pre-training, they fine-tune those models on GLUE and SuperGLUE, and the inputs during fine-tuning only contain some characters of the original word, as in pre-training. They find that even when only keeping a few ($\leq 3$) characters of the word, the PLM still performs quite well on downstream tasks. Probing experiments reveal that even when most characters are dropped from a word, the hidden representation of the PLM still encodes some linguistic information.

Contribution types
===
Computationally-aided linguistic analysis, NLP engineering experiment

**Questions For The Authors:**

A. For Vowels (V), what happens if a word contains no vowel?

B. Is there any explanation for why L has the lowest pre-training loss?

C. I would like to see the majority baseline for the probing experiments.

**Reasons To Accept:**

1. The paper has a clear takeaway, and the topic is never explored before. They show that we can pre-train a PLM that still works reasonably on downstream tasks even when the input tokens contain a few characters.
2. The paper is well-written

**Reasons To Reject:**

1. Some parts of the paper are difficult to understand. Precisely, the whole Section 6.2 is unclear to me when I first read it. The phrase "`cut all words to keep the threshold`" is odd. I thought what this paper does is dropping characters from a word, but "`cut all words`" sounds like you are removing words from the sentences that are longer than the threshold.
2. The results of the paper may not be directly related to how current PLMs learn, so it does not really push our knowledge of existing PLMs a step further. The results in this paper can only tell us PLMs can still learn something when input characters are dropped. However, since existing PLMs do not pre-train or fine-tune on character-dropped datasets, we still do not know if PLMs trained on full tokens can comprehend character-dropped inputs. The experiments in this paper are quite different from the cognitive science motivation. In cognitive science, the takeaway is that humans learning from full tokens can understand partial words, but the experiments in this paper show that models pre-train on partial words can comprehend partial words. To address this issue, a simple experiment is fine-tune the full token model on downstream tasks whose inputs are {F,M,L, FF, FL, etc.}.
3. The results of the probing experiments are not as surprising as the paper states. This is because when keeping the first, middle, and last characters (FML) in the sentence, a lot of information for syntactic and semantics information is kept. This is because many pronouns are less than three characters (e.g., he, she, it, we), so FML does not really drop any characters in those words. This is also the case for the verb be (e.g., is, are, was) and auxiliary (do, did). I think the papers should conduct a more detailed analysis of the error pattern of the probing experiments. Is the probing accuracy higher for sentences whose verbs and pronouns are not character-dropped?



**Reproducibility:**

5: Could easily reproduce the results.

**Reviewer Confidence:**

5: Positive that my evaluation is correct. I read the paper very carefully and I am very familiar with related work.

**Typos Grammar Style And Presentation Improvements:**

- Words in the Figures in the Appendices are too small to read.
- Line 361: trained **on** partial tokens

---

> ### Author Rebuttal · Authors · 2023-08-28
>
> Thank you so much for your constructive feedback.
>
> 1. Regarding Section 6.2, we are going to rewrite the section to make it clearer as suggested. “cut all words to keep the threshold” means that we keep only the specified characters for each token and remove any other characters if they exist.
>
> 2. In our experiments, we pre-train two types of models using two different pre-training objectives. The first one is to pre-train on partial tokens and predict the same partial tokens. The second one is to pre-train on partial tokens and predict the full tokens. We agree that this is slightly different from the cognitive science motivation but we consider it as a more simple setting that allows us to explore the predictive capacity of LLMs under information loss. Thank you for the suggestion to fine-tune the full token model on downstream tasks using partial tokens {F, M, L, FF, FL, etc.}. This is really an interesting experiment and it is easy to implement. Intuitively, we expect the full token model to achieve very good results when fine-tuned on downstream tasks with partial input tokens. We are going to include this and report the results in the camera-ready version of the paper.
>
> 3. We agree that when keeping three characters (FFF, FML and LLL) of the input tokens, some pronouns and auxiliary verbs do not suffer any information loss. However, many other verbs and nouns contain more than three characters and therefore suffer from information loss. In addition, the probing tasks we utilized in our paper cover a number of syntactic and semantic information tasks such as SubjNum which predicts whether the subject of the main clause is singular or plural. The suggestion to include error analysis for the probing experiments sounds interesting. We can address that in the camera-ready given the extra space.
>
> **Response to Q A:** For Vowels (V) models, if a word contains no vowel, it will be dropped completely from the sentence. However, this is very rare as the vast majority of words in English contain vowels. We will better clarify this in the camera-ready.
>
> **Response to Q B:** This is most likely due to the initialization processes as all three models (F, M and L) have the same architecture and pre-training setup.  The combination of starting parameter values and random initialisation might help the L model achieve a slightly lower pre-training loss compared to the F and M models. However, when we finetune the L model on downstream tasks we observe that it achieves the lowest average score compared to the F and M models. We will make this more clear in the discussion of the results.
>
> **Response to Q C:** Thank you for the suggestion. The majority baseline for the probing experiments will be added in the camera-ready version of the paper.
>
> Thank you for pointing out the missing references for each dataset in GLUE and SuperGLUE. This will be addressed in the camera-ready version.
>
> The presentation of the Figures in the Appendices will be enhanced in the camera-ready version. Thanks for the suggestion.

---

### Official Review · Reviewer_tAA1 · 2023-07-30

**Typos Grammar Style And Presentation Improvements:** It would be helpful to show % of base…
**Soundness:** 5

**Excitement:**

4: Strong: This paper deepens the understanding of some phenomenon or lowers the barriers to an existing research direction.

**Paper Topic And Main Contributions:**

The authors pretrain BERT-base on English corpora with different schemes to trim the input vocabularies, e.g., representing every word with its first character, or just the last character, or both, or first-middle-last, or only vowels, or only consonants, etc. This is inspired by studies showing how humans can read very well with partially boldened words, even with mispellings.

They pretrain and finetune each model in two settings: (1) the output is also collapsed into the resulting vocabulary from the corresponding changes to input, or (2) the output vocabulary is left intact. Their baseline is not BPE or WordPiece but a word-level BERT which includes all words in the vocabulary.

Their major contribution is to show that despite losing a lot of the information by replacing words with only a subset of characters (as few as one character per word), they can achieve close to baseline performance, not only on MLM intrinsic task but also on GLUE and SuperGLUE (even the original task setting, where the output vocabulary stays same).

They also show how first characters are more informative than the middle or last, and that their variants capture more syntactic information than even the baseline.

**Questions For The Authors:**

A) Are there exactly 50,010 unique words in the corpus or is this truncated somehow? What happens to words not seen in pretraining but seen in finetuning on GLUE/SuperGLUE?
B) Do any of GLUE/SuperGLUE tasks have different vocabularies in output space or are they all classification tasks like NLI? If the former, then are not most comparisons unfair, similar to how comparing perplexities across tokenizers with different vocabularies is unfair?

**Reasons To Accept:**

Despite the proposed variants not occurring naturally (nor obviously outperforming the baseline), this is an interesting experiment that sheds light on how English language non-uniformly (first > middle, last) and redundantly, which can mostly be comprehended even when most of the characters are removed.
One way to interpret this paper is as a simulation of several (below average) human learners (i.e., BERTs) being trained to learn English and take tests (GLUE/SuperGLUE) but with access to only a small set of characters in each word.

Alternatively, their stance of "testing the limits of PLMs of learning with partial information" is also a worthwhile research question for which we find valuable answers to, albeit only on English.

**Reasons To Reject:**

I contest the fairness of the evaluation leading to their "main results" since the output space is different for different settings on GLUE/SuperGLUE where the answer is natural language.
Thankfully, the corresponding fair results are in the Appendix (the setting where output vocabulary is kept constant).

Their claims are exaggerated to properties "natural language" in general, even though the experiments are only on English.

The motivation is not very strong, the reader may be left wondering "so what"? Discussion section can be expanded to consider practical downstream (even if far fetched) ways to exploit the results observed in these experiments.

**Reproducibility:**

5: Could easily reproduce the results.

**Reviewer Confidence:**

4: Quite sure. I tried to check the important points carefully. It's unlikely, though conceivable, that I missed something that should affect my ratings.

---

> ### Author Rebuttal · Authors · 2023-08-28
>
> Thank you so much for your constructive feedback.
>
> We experimented only with English due to limited computational resources. Expanding our experiments to cover other languages is an important avenue for future work.
>
> We appreciate your suggestion to enhance the Discussion section by considering the practical downstream tasks that can make use of the outcomes observed in our experiments. We will extend the Discussion section based on the results observed given the extra space in the camera-ready version.
>
> **Response to Q A:** This is truncated as we use the most frequent 50,000 unique words in addition to 10 tokens for punctuation marks and special tokens.
> Any words not in the vocab are replaced with the unknown “<unk>” special token.
>
> **Response to Q B:** The GLUE/SuperGLUE benchmarks cover different types of tasks, and they exhibit varying characteristics in terms of the output space vocabulary. While some of these tasks are indeed classification tasks similar to NLI, there are also tasks that involve different output vocabularies. We will bring the full results from the appendix to the main body of the paper, as you suggested, given the extra space in the camera-ready version.
>
> Thank you for the suggestion to show the majority class baselines. We will add them in all tables given the extra space in the camera-ready version.

---

### Official Review · Reviewer_77VM · 2023-08-07

**Soundness:** 3

**Excitement:**

3: Ambivalent: It has merits (e.g., it reports state-of-the-art results, the idea is nice), but there are key weaknesses (e.g., it describes incremental work), and it can significantly benefit from another round of revision. However, I won't object to accepting it if my co-reviewers champion it.

**Missing References:**

1. Madasu, A. and Srivastava, S., 2022, December. What do Large Language Models Learn beyond Language?. In Findings of the Association for Computational Linguistics: EMNLP 2022 (pp. 6940-6953).

2. Wu, Yuhuai, Felix Li, and Percy S. Liang. "Insights into pre-training via simpler synthetic tasks." Advances in Neural Information Processing Systems 35 (2022): 21844-21857.

**Paper Topic And Main Contributions:**

What is this paper about:
This paper is poses an interesting question about LMs: Do they need all the characters in a word to understand the meaning? To explore this, the authors pre-train BERT with one two and three characters of the entire word. The different pre-trained models are then fine-tuned on GLUE, SuperGLUE and linguistic tasks to evaluate their performance.


Main Contributions:
Focuses on important questions: This paper explores an interesting question related to language models: How much information in a word is required for LMs?

Sufficient methodology: The methodology introduced in the paper is sufficient to understand the problem this paper deals with.

Usefulness to the NLP community: The findings that LMs perform well even with single and two characters of a word is an interesting finding to the community.

**Reasons To Accept:**

Reasons to Accept:

Novelty: This paper studies a novel (insightful) question that is quite relavant to the NLP community.

Supporting experiments: The authors did an excellent job at performing supporting experiments to explore the proposed study.

Well written paper: The paper is well-written and easy to follow.

Future improvements suggestions:
1. If possible, the authors can also experiment with the same protocol on downstream tasks i.e using single, two and three characters and measure the performance on them. This can bring out additional insights.

2. Smaller sizes of LMs: The authors can also study for smaller sizes of LMs and evaluate how much size of a LM matter.

**Reasons To Reject:**

There are some weaknesses that are major (needs additional experiments) weaknesses.

Generalization: The authors experiment with just BERT for the proposed study on LMs. Findings on one model may not be used to generalize across all the LMs. Although the authors argue this in limitations, there is a work around for this. Instead of using a big corpus like BookCorpus, the authors can pre-train on a subset of this data which wouldn't be computationally costly. In addition, the authors may consider smaller datasets used in [1].

**Reproducibility:**

4: Could mostly reproduce the results, but there may be some variation because of sample variance or minor variations in their interpretation of the protocol or method.

**Reviewer Confidence:**

5: Positive that my evaluation is correct. I read the paper very carefully and I am very familiar with related work.

---

> ### Author Rebuttal · Authors · 2023-08-28
>
> Thank you so much for your constructive feedback.
>
> 1. **Experimental protocol:** Thank you for the suggestion. The same protocol used for pre-training is also used on downstream tasks for each model. For example, when fine-tuning the first character model (F), we use the first character of each token in the downstream tasks. We are going to include extra experiments in the camera-ready version where we fine-tune the full token model on downstream tasks using partial input tokens (i.e. using single, two and three characters).
>
> 2. **LMs of smaller sizes:** This is an interesting idea and can be explored in future work.
>
> Regarding the experiment with other LM architectures, we agree that our experiments can be extended to cover other LM models such as GPT in future work. The suggestion of using smaller datasets to pre-train other models sounds interesting. We will consider that in our future work.
>
> Thank you for pointing out the missing references. We will add them in the camera-ready version of the paper.

---

### Meta-Review · Area_Chair_xbt5 · 2023-09-18

**Recommendation:** 5

**Metareview:**

This paper explores how robust language models are to missing input token characters. Their results are surprising: they find that language models retain high performance when pre-trained with extreme character loss. The reviewers agree that this is a novel research question, and, on the whole, that the methodology is sound. There is a concern about the reliability of the results given that only BERT is studied, but BERT remains a very commonly used model (particularly in computational humanities and social science), so I think this concern is outweighed by the computational costs of additional experiments, as acknowledged by the reviewer who raised this concern. I do agree that it is important to revise the text to clarify the scope of the findings (English).

---

### Decision · Program_Chairs · 2023-10-07

**Decision:**

Accept-Main

**Comment:**

This paper explores how robust language models are to missing input token characters. Their results are surprising: they find that language models retain high performance when pre-trained with extreme character loss. The reviewers agree that this is a novel research question, and, on the whole, that the methodology is sound. There is a concern about the reliability of the results given that only BERT is studied, but BERT remains a very commonly used model (particularly in computational humanities and social science), so I think this concern is outweighed by the computational costs of additional experiments, as acknowledged by the reviewer who raised this concern. I do agree that it is important to revise the text to clarify the scope of the findings (English).